# Optimized Protocol for In Vitro Pollen Germination in Yam (*Dioscorea* spp.)

**DOI:** 10.3390/plants10040795

**Published:** 2021-04-18

**Authors:** Jean M. Mondo, Paterne A. Agre, Robert Asiedu, Malachy O. Akoroda, Asrat Asfaw

**Affiliations:** 1International Institute of Tropical Agriculture (IITA), Ibadan 5320, Nigeria; m.mubalama@cgiar.org (J.M.M.); r.asiedu@cgiar.org (R.A.); a.amele@cgiar.org (A.A.); 2Institute of Life and Earth Sciences, Pan African University, University of Ibadan, Ibadan 200284, Nigeria; 3Department of Crop Production, Université Evangélique en Afrique (UEA), Bukavu 3323, Democratic Republic of the Congo; 4Department of Agronomy, University of Ibadan, Ibadan 200284, Nigeria; malachyoakoroda@gmail.com

**Keywords:** Brewbaker and Kwack medium, *D. alata*, *D. rotundata*, pollen viability and storage, in vivo fertilization

## Abstract

Yam (*Dioscorea* spp.) plants are mostly dioecious and sometimes monoecious. Low, irregular, and asynchronous flowering of the genotypes are critical problems in yam breeding. Selecting suitable pollen parents and preserving yam pollen for future use are potential means of controlling these constraints and optimizing hybridization practice in yam breeding programs. However, implementing such procedures requires a robust protocol for pollen collection and viability testing to monitor pollen quality in the field and in storage. This study, therefore, aimed at optimizing the pollen germination assessment protocol for yam. The standard medium composition was stepwisely modified, the optimal growth condition was tested, and in vivo predictions were made. This study showed that the differences in yam pollen germination percentage are primarily linked to the genotype and growing conditions (i.e., medium viscosity, incubation temperature, and time to use) rather than the medium composition. The inclusion of polyethylene glycol (PEG) in the culture medium caused 67–75% inhibition of germination in *D. alata*. Although the in vivo fertilization was dependent on female parents, the in vitro germination test predicted the percentage fruit set at 25.2–79.7% and 26.4–59.7% accuracy for *D. rotundata* and *D. alata* genotypes, respectively. This study provides a reliable in vitro yam pollen germination protocol to support pollen management and preservation efforts in yam breeding.

## 1. Introduction

The yam is a multispecies crop belonging to the genus *Dioscorea* and provides food, medicine, and income in tropical and subtropical areas of America, Africa, Asia, the Caribbean, and Oceania [1]. Of the ~600 yam species, white yam (*D. rotundata* Poir.), yellow yam (*D. cayenensis* Lam.) and water yam (*D. alata* L.) represent ~95% of the global yam production [2]. Six African countries, namely Nigeria, Ghana, Côte d’Ivoire, Benin, Togo and Cameroon, account for more than 90% of the annual global yam production [3]. In these countries, yam is not only a food and income security crop; it is also an integral part of the people’s socio-cultural and religious belief systems.

Most popular yam species are predominantly dioecious, with male and female flowers borne on separate individual plants. Indeed, individuals of a yam cultivar are unisexual (being either male or female) although cases of monoecious plants (individuals with both male and female flowers) exist [4,5,6]. Developing new yam varieties requires, therefore, hybridization among selected parents to create genetic variability and selection of the progenies’ superior clonal derivatives for release after a series of testing. Successful hybridization depends primarily on the pollen germination ability (viability) of the male parent and the stigma receptivity of the female under favorable weather conditions [6]. In this work, two aspects of pollen viability are discussed: the pollen’s germination ability tested in the laboratory, and its ability to fertilize a female flower after hand pollination [7].

Yam hybridization activities are often constrained by low, irregular and asynchronous flowering [5,6]. Selecting suitable pollen parents and preserving yam pollen for future use are potential means of managing these constraints [6,8]. However, implementing these practices in yam breeding programs requires a cost-effective and reliable pollen viability testing protocol, allowing the discrimination of genotypes for viability status and efficient monitoring of stored pollen.

Past studies on yam pollen used viability (germination) testing protocols from other crops without prior optimization [8,9,10], which led to a low uptake by the breeding programs. There is, therefore, a need to develop an optimized pollen viability testing method for yam to increase pollination efficiency, facilitate efficient monitoring of stored pollen and for genetic and physiological studies. Mondo et al. [6] reviewed the most popular pollen viability testing methods and their potential in improving pollination efficiency in yam breeding programs. These include in vitro germination, the vital staining of pollen grains, in vivo fertilization and seed development, and impedance or optical flow cytometry. In vitro germination method consists of assessing the germination of pollen on a nutritive medium. This method is highly correlated with in vivo pollination for most crops. For instance, Ng and Daniel [10] established a positive relationship between in vitro germination and fruit set after hand pollination of white Guinea yam (*D. rotundata*). That positive relationship was also confirmed on other plant species by Volk [11]. Mondo et al. [6] suggested that the Brewbaker and Kwack [12] culture medium previously used in yam pollen studies should be optimized for the concentration of the boric acid, calcium nitrate, sucrose and the pH as requirements vary with species. The effects of other factors such as the type of medium (semi-solid or liquid), germination temperature, incubation duration and the inclusion of polyethylene glycol in the medium on pollen germination have not been determined for yam and should, therefore, be investigated for the optimum in vitro germination testing.

This work aimed at establishing a pollen germination assessment protocol by determining the optimal germination medium composition for yam pollen and suitable growth conditions.

## 2. Results

The development of the pollen germination protocol for yam was achieved in three steps. We first identified the optimum growing conditions after assessing effects of incubation duration and temperature, medium viscosity (levels of the agar concentration), and delay in the use of collected pollen. Best growing conditions were then used in optimizing the medium composition through a series of single-factor experiments. Lastly, we assessed the prediction ability of the optimized protocol vis-à-vis in vivo fertilization using a series of hand pollinations.

### 2.1. Growth Conditions and Pollen Germination Percentage

Increasing the time interval between pollen collection and culture negatively affected the pollen germination ability (*p* < 0.001). About 50–75% germination ability was lost in 3–4 h after pollen collection from the field, regardless of the genotype (Figure 1). The agar concentration and the genotype significantly affected the *D. rotundata* pollen grain germination (*p* < 0.001). The highest mean germination levels were recorded on 0.5% (14.2% germination) and 1% (13.5% germination) of agar, while the lowest (4.9%) was on the hanging drop without agar. TDr1621001 pollen grains germinated at a higher rate (17.4%) than those of TDr1655018 (4.7%) (Table 1). Other factors, such as the incubation temperature and duration had no significant effects on germination percent. However, the interactions genotype × temperature (*p* = 0.02), genotype × agar (*p* < 0.001), temperature × agar (*p* = 0.005), and genotype × temperature × agar (*p* < 0.001) were significant.

*Dioscorea alata* pollen germination was sensitive to the temperature (*p* = 0.034), the agar concentration (*p* < 0.001) and the genotype effects (*p* < 0.001). The best results were recorded on the medium with 0.5 and 0.75% agar (14.9 and 11.3% germination, respectively). The liquid medium (without agar) had the lowest germination (Table 2) consistently. Pollen grains from TDa1662006 germinated better than those from TDa1662010 (12.7 vs. 6.6%). The optimum pollen germination temperature for *D. alata* was 25 °C.

### 2.2. Medium Composition and Pollen Germination

In all the single-factor experiments (Figure 2, Figure 3, Figure 4 and Figure 5), the percent pollen germination differences were exclusively under the genotype effects for both yam species. For *D. rotundata*, pollen of TDr1621012 always had higher percent germination than TDr1655018 (25.2–44.0 vs. 6.2–13.2%). The trend was similar for *D. alata*; pollen of TDa1662006 had consistently higher percent germination (31.5–39.6%) compared to TDa1662010 (9.8–11.2%).

### 2.3. Effects of Polyethylene Glycol on Yam Pollen Germination

The best single-factor experiments’ composition media were selected to evaluate the effects of polyethylene glycol (PEG) on percent yam pollen germination. For *D. rotundata*, the medium composition at this stage was: 12.5% sucrose, 125 ppm boric acid, 375 ppm calcium nitrate, and adjusted at pH 6.5. On the other hand, the medium composition for *D. alata* was made of: 5% sucrose, 100 ppm boric acid, 450 ppm calcium nitrate, and adjusted at pH 7. All other medium ingredients (MgSO_4_.7H_2_O and KNO_3_) not optimized in this study were kept at the same concentrations as in the Brewbaker and Kwack (BK) medium [12]. At this stage, 0.5% agar, 25 °C and the incubation duration of 3 h were used for both species.

The incorporation of PEG in the optimized medium as well as the interaction genotype × PEG had no significant effects on *D. rotundata* pollen germination percent (Table 3). However, significant effects of the genotype (*p* = 0.004), PEG application (*p* = 0.02) and the interaction PEG × genotype (*p* = 0.04) were recorded for *D. alata*. The presence of PEG in the medium significantly reduced *D. alata* pollen germination ability by 67–75% (Table 3).

Comparing the optimized media with the starting Brewbaker and Kwack (BK) medium showed no significant differences regardless of the genotype and species. We also discovered that the established culture medium for *D. rotundata* was unsuitable for *D. alata*. For instance, the *D. alata* germination rate was significantly lower when grown on the medium optimized for *D. rotundata* (7.1%) compared to its performance on the medium optimized for *D. alata* (24.5%) and the BK medium (20.0%).

### 2.4. Correlations Between In Vitro Germination and In Vivo Fertilization

To assess the accuracy of the in vitro pollen germination in predicting fruit set, we added to the four genotypes in the previous steps, other pollen parents which were readily available at the time of this validation step. We found that although the female parents’ cross-compatibility highly influenced the in vivo fertilization, the in vitro germination test predicted the percentage fruit set at 25.2 to 79.7% and 26.4 to 59.7% accuracy for *D. rotundata* and *D. alata* genotypes, respectively (Table 4). The schematic presentation of the optimized protocol is provided in Figure 6.

## 3. Discussion

A reliable pollen germination testing method is crucial for improving the breeding practices and the pollination efficiency in the low and asynchronous flowering yam plant. The literature shows that pollen germination depends on species, cultivar, and growing conditions [13]. The present study assessed the effects of growing conditions and medium composition on the pollen germination of *D. rotundata* and *D. alata* genotypes. Based on this study, the collected pollen should be used within 2 h for maximum results, as the viability is significantly reduced after 3–4 h. The screening of a large number of genotypes will be challenging as it takes 8–12 min for culturing pollen in a single Petri dish. Moreover, the recommended time for yam pollen collection is limited to the morning hours (8–11 a.m.) [8]. Rapid viability testing methods will, therefore, be necessary if high throughput phenotyping is desired for yam pollen. The pollen germination protocol optimized in this study will serve as the basis for calibrating other methods such as staining, impedance and optical flow cytometry as well as automated pollen count [6].

The extent of in vitro pollen germination was primarily associated with the genotype and growing conditions rather than the medium composition for both yam species. This finding ascertained the suitability of the standard Brewbaker and Kwack (BK) medium for yam species without optimizing the sucrose, calcium nitrate and boric acid. The BK medium was successfully tested on 86 species from 79 genera and 39 families [12]. This study also endorsed 25 °C as the optimal germination temperature for yam species, as it was also recommended on most plant species [11]. However, the *D. rotundata* had an extensive germination temperature range (15–35 °C) compared to *D. alata* genotypes. The finding from this study suggests using the semi-solidified medium rather than the liquid (hanging drop without agar) medium. Silva et al. [14] showed that the agar plays several roles in the germination medium: it promotes the solidification and the osmotic equilibrium of the culture medium, ensures a constant relative humidity, and facilitates the incorporation of nutrients, aiding the formation of the pollen tube. In opposition to previous yam pollen studies which used 0.7% agar, this study found consistently better results with 0.5% agar. This concentration may have favored high water and nutrient absorptions for the pollen tube’s optimal formation and growth [14]. The absence of full information on the agar type, autoclaving, and handling procedures used in previous pollen studies on yam makes it difficult to conclude whether observed positive results with 0.5% agar are only attributed to the agar concentration or other handling procedures. Our results supported Daniel et al. [8]’s method for the 3 h incubation duration as no significant differences were found with the 18 h incubation duration used by Akoroda [9]. Although some yam pollen grains germinated after 3 h, the initial trends remained unchanged at 18 h. This study does not recommend using polyethylene glycol (PEG) on culture medium as it reduced the pollen germination percent.

Unlike Akoroda [9] and Daniel et al. [8], the in vitro germination percentage was consistently higher than the in vivo fruit set for *D. rotundata*. Several environmental factors (rainfall, relative humidity, temperature, sunshine, etc.), stigma receptivity and cross-compatibility barriers affect the successful pollination and fruit set under field conditions, although viable pollen was used [6,7]. These factors often result in low correlations between laboratory estimates and in vivo fertilization data. Previous studies on yam found no clear trend in predicting field fruit set using in vitro germination estimates, probably because the used medium and growing conditions, adapted from other crop species, were not suitable for optimum yam pollen germination [8,9,15]. Our study showed a positive correlation trend between pollen germination percent and the fruit set in field after hand pollination for both *D. rotundata* and *D. alata* genotypes. It was, thus, possible to discriminate the genotypes based on pollen viability status using in vitro testing data. The positive relationship between in vitro germination and in vivo fruit set was also reported by a study on yam [10] and other plant species [11].

Although our protocol could rank the genotypes, determining the exact germination percents was mostly under the weather conditions’ effects before and during the pollen collection. This could explain fluctuations in genotype means without affecting the ranking. Compared to the pollen germination ranges from the yam literature (0.3–85% for *D. rotundata* and 20–98% for *D. alata* [6]), the in vitro germination percent in this study seldom surpassed 45%, regardless of the species and treatment. This could be attributed to the hybrid nature of plant materials used in this study. Alexander [16] observed reduced pollen viability of hybrids compared to either parent pollen in ornamental plants.

## 4. Materials and Methods

### 4.1. Plant Material

Two genetically known and profusely flowering male genotypes of *D. rotundata* (TDr1621012 and TDr1655018) and two of *D. alata* (TDa1662010 and TDa1662006) were selected for this study. They were all breeding lines and diploids. After flowering has ceased for the two *D. rotundata* genotypes, they were replaced by relative genotypes (TDr1621003 and TDr1614205) to complete the experiment. Past experiences have shown that the pollen of yam genotypes from same crossing families tended to behave alike. Female parents readily available at the in vivo fertilization step were sourced from yam crossing blocks established in April 2020 on a research field at the International Institute of Tropical Agriculture (IITA), Ibadan, Nigeria (7°29′ N and 3°54′ E). The list of female genotypes used in this study is presented in Table 5. They were all breeding lines, diploids and with good crossability history. Optimum crop husbandry practices such as remoulding of ridges, timely weeding, training of vines on stakes, and irrigation were applied in the crossing blocks, as needed, during the growing season. No fertilizer was applied.

### 4.2. Male Flower Harvest and Pollen Extraction

In Nigeria where IITA yam crossing blocks are established, the flowering window extends from July–September and September–November for *D. rotundata* and *D. alata* species, respectively. Pollen germination testing was, thus, conducted in those time intervals. On the day of pollen germination experiments, recently opened male flowers were collected between 8:30 and 9:30 a.m. when the weather condition was still cool. Flowers were harvested and kept in well-labeled plastic vials and deposited in a basket for transport. Since the laboratory was close to crossing blocks and the weather was still cool, no particular preservation measure (in terms of temperature or humidity) was taken during transport from the field to the laboratory.

Except for the experiment testing the effect of delay between pollen collection and culture, the harvested pollen was used within 2 h after collection from field. Due to the tiny size and the sticky nature of yam pollen grains, anthers were removed from the flower buds and deposited on the medium to release pollen. A binocular magnifying glass and a pin facilitated the anther removal and transfer. Since not all yam anthers contain pollen, a total of 50 anthers were deposited in each Petri dish to ensure availability of sufficient pollen grains for scoring.

### 4.3. Assessment of the Optimal Growth Conditions for In Vitro Pollen Germination

Growth conditions tested were mainly the effect of delay in pollen use, incubation temperature and duration, and the medium viscosity. For this experiment, we used the Brewbaker and Kwack (BK) medium (i.e., 10% sucrose, 100 ppm H_3_BO_3_, 300 ppm Ca(NO_3_)_2_.4H_2_O, 200 ppm MgSO_4_.7H_2_O, 100 ppm KNO_3_ in distilled water) [12]. The pH was adjusted to 6.5 by adding HCl (acid) or NaOH (base) to the medium using a magnetic stirrer and a digital pH-meter. Solutions were then autoclaved for 15 min (121 °C, 1 atm.), cooled for 30 min, pipetted into 60 mm plastic Petri dishes and stored at 4 °C in a refrigerator for use by the next day.

To test the effect of the medium viscosity, two types of medium were prepared; the liquid or hanging drop without agar and the semi-solidified medium with three agar levels (0.5%, 0.75% and 1%). The agar used in this study is of the Glentham Life Sciences Trademark (Corsham, United Kingdom, (C_12_H_18_O_9_)_n_). Effects of the delay in use of collected pollen were assessed using five time intervals: <1 h, 1–2 h, 2–3 h, 3–4 h, and 4–5 h. Sufficient male flower sample was collected once in the morning and kept on the laboratory bench at ambient conditions (~25 °C). The incubation temperature and duration effects were evaluated by incubating cultivated Petri dishes for 3 and 18 h under 15, 25 and 35 °C (we used simultaneously three different incubators).

### 4.4. Medium Composition Optimization

After determining the optimal medium type (liquid or semi-solidified), agar concentration, incubation duration and temperature, and the best time interval between pollen collection and use; the medium composition was then optimized. As suggested by the literature, the single-factor optimization experiments for medium composition focused on boric acid, calcium nitrate, sucrose and pH. The Brewbaker and Kwack medium (BK) was used as the starting medium and was step-wisely modified to find the optimum media for *Dioscorea* spp. Boric acid (50, 75, 100, 125 and 150 ppm), calcium nitrate (150, 225, 300, 375 and 450 ppm) and sucrose (5%, 7.5%, 10%, 12.5% and 15%) were tested at 50%, 75%, 100%, 125% and 150% of the initial concentrations in the Brewbaker and Kwack (BK) medium.

Culture media with optimized concentrations were then tested at different pH (5.5, 6.0, 6.5, 7.0 and 7.5). At the end of the medium optimization process, the effects of polyethylene glycol (PEG 8000, Sigma Chemical Co, St. Louis, MO, USA) at a concentration of 60 g L^−1^ were assessed. PEG was applied on the medium after autoclaving but before the complete solidification of the medium. Once the PEG was added to the medium, the flask was shaken gently to ensure homogeneity (equal distribution) of the solution before the medium solidified. It was, thus, applied on the medium after autoclaving but before pipetting in Petri dishes.

As for the previous steps, solutions were autoclaved for 15 min (121 °C, 1 atm.), cooled for 30 min, pipetted into 60 mm plastic Petri dishes and stored at 4 °C in a refrigerator for use by the next day. The Petri dishes containing pollen were sealed prior and during incubation using a parafilm tape. The relative humidity in the incubator was not recorded. The pollen germination was in dark for all experiments (used incubators had light bulb but which automatically switched off upon closure of the incubator solid outer door).

### 4.5. Pollen Germination Scoring

After the incubation, a fluorescence microscope (Olympus BX51, Tokyo, Japan) was used to visualize at 10× magnification and to count the germinated pollen grains from each anther manually. As suggested by Dafni and Firmage [7], pollen grains were considered germinating when their tubes elongated to a length that was at least their diameter.

### 4.6. In Vivo Fertilization

To determine the correlation between the in vitro germination and in vivo fertilization methods, the average performance of each pollen parent in crosses with distinct female genotypes was recorded. At flowering, female flowers were bagged with thrips-proof cloth bags 5 days before pollination. Hand pollination was then carried out and pollinated flowers were bagged for two weeks. At the validation step, male flowers used for pollen germination in the laboratory were sourced from the same individuals providing pollen for hand pollination in the field. The number of female flowers pollinated in a cross-combination depended on the availability of either mature male or receptive female flowers on the day of crossing. Moreover, some female genotypes produced more flowers than others, and thus, the number of flowers pollinated per female genotype was not identical for all involved females. Due to existing cross-compatibility barriers in yam breeding, cross-combinations were decided based on crossability history of both male and female genotypes. All the crosses were performed in the morning hours (8 a.m. to 12 noon). Successful fruit set was evaluated two weeks after hand pollination. It consisted of opening the cloth bags and manually counting the number of fruits developed on a spike. The percentage fruit setting rate was then calculated as follows:(1)$$Fruit\;setting\;rate\;\left(\%\right)=\frac{Number\;of\;fruits\;set}{Number\;of\;flowers\;pollinated}\times 100$$

### 4.7. Data Analysis

Experiment on testing effects of delay in culture after pollen collection from the field had five replications and 50 pollen grains scored per replication, making a total of 250 pollen grains scored per time interval. After discovering that a yam pollen sample was significantly losing viability after 2 h, we reduced the number of replications for the following steps to ensure pollen culture for every experiment was completed within 2 h after pollen collection from the field. Therefore, all the laboratory experiments for testing pollen growing conditions and the medium composition optimizations were conducted in two replications for each treatment. A total of 50 pollen grains were scored per replication, making the overall number to 100 pollen grains scored for each treatment. It is noteworthy that yam pollen is sticky to the anther and it was requiring 8–12 min for extracting and culturing anthers with pollen in a single Petri dish.

Data on the effect of delay in pollen culture, medium composition and growth condition optimizations were subjected to variance analysis and the means were separated by Fisher’s least significant difference (LSD) test at 5% *p*-value threshold, using the lme4 package implemented in R software [17]. The in vitro germination test’s prediction accuracy was calculated by comparing the germination percent (expected value) and the overall in vivo fruit set percent (observed value) for each male parent in a series of hand pollinations.

## 5. Conclusions

This study showed that the yam pollen germination is primarily influenced by the genotype and the growing conditions such as the incubation temperature, the medium viscosity and time to use. The Brewbaker and Kwack’s medium used in previous yam pollen studies is still suitable without optimizing the boric acid, calcium nitrate, sucrose and pH. This protocol provides a possibility of ranking genotypes and predicting the in vivo fruit set using in vitro pollen germination information. It is, therefore, filling the gap which existed in yam breeding by providing an optimized protocol for in vitro testing of yam pollen germination. It will also serve as the basis for calibrating rapid pollen viability testing methods to allow high throughput pollen phenotyping for yam breeding programs.

## Figures and Tables

**Figure 1 plants-10-00795-f001:**
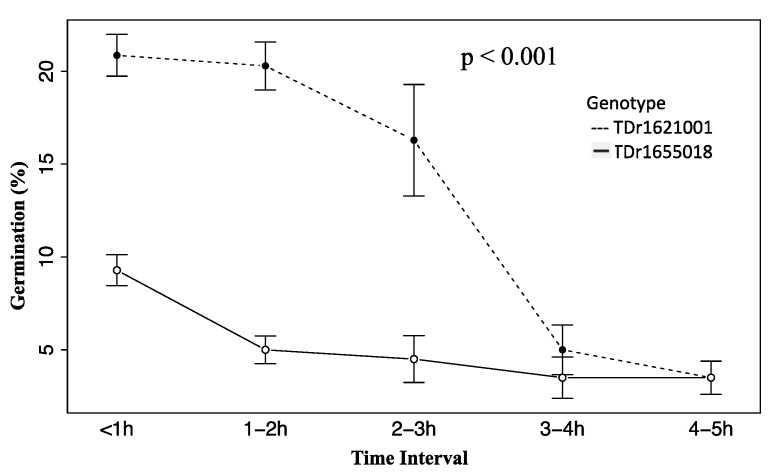
Effect of the delay between pollen collection and culture on percentage germination of two *D. rotundata* genotypes (*p* < 0.001).

**Figure 2 plants-10-00795-f002:**
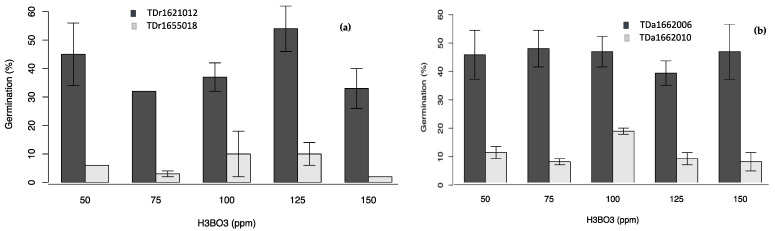
Effects of boric acid on pollen germination of genotypes of (**a**) *D. rotundata*, (**b**) *D. alata.*

**Figure 3 plants-10-00795-f003:**
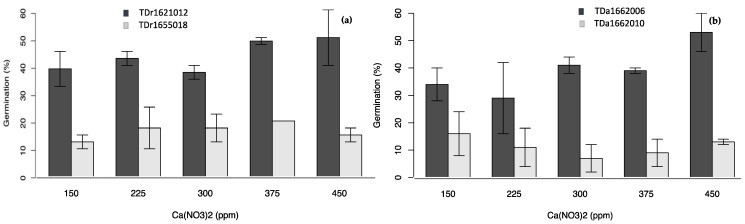
Effects of calcium nitrate on pollen germination of genotypes of (**a**) *D. rotundata*, (**b**) *D. alata.*

**Figure 4 plants-10-00795-f004:**
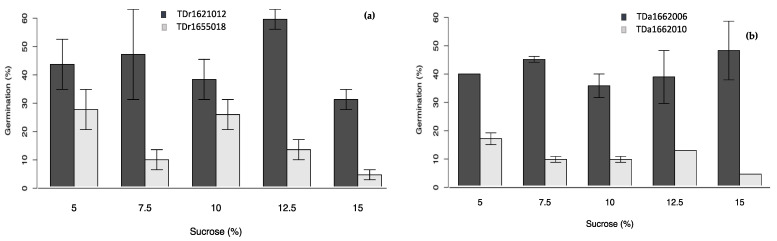
Effects of sucrose on pollen germination of genotypes of (**a**) *D. rotundata*, (**b**) *D. alata.*

**Figure 5 plants-10-00795-f005:**
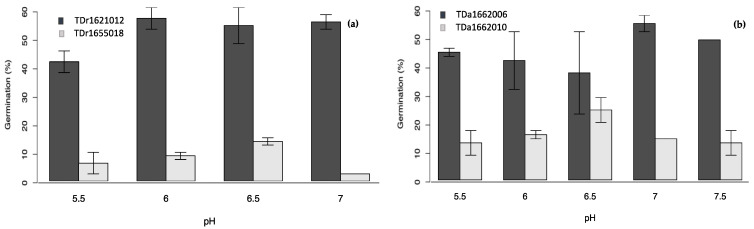
Effects of the medium pH on pollen germination of genotypes of (**a**) *D. rotundata*, (**b**) *D. alata.*

**Figure 6 plants-10-00795-f006:**
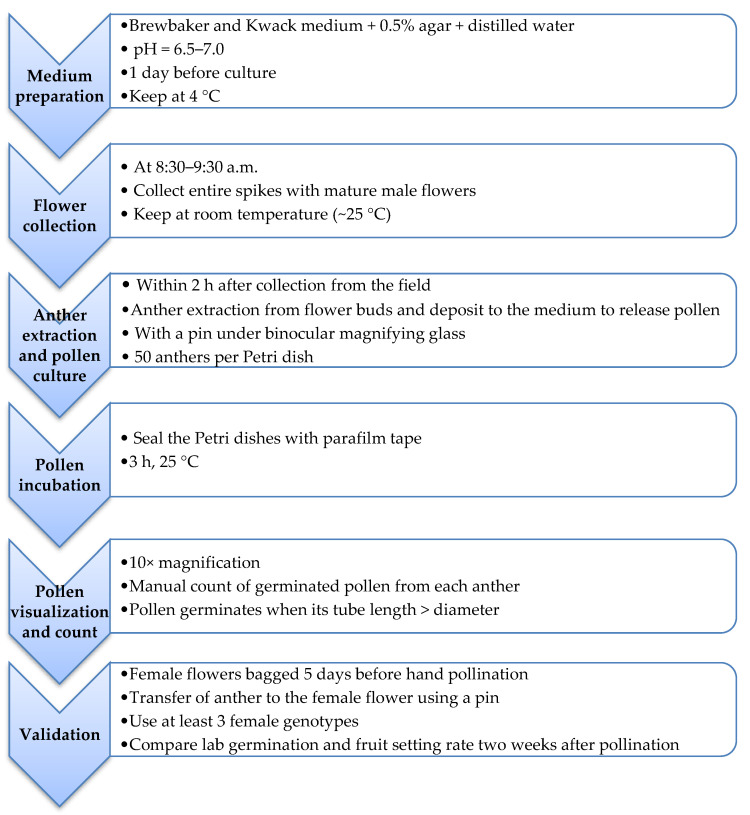
Schematic presentation of the optimized protocol for in vitro pollen germination testing in *Dioscorea* spp.

**Table 1 plants-10-00795-t001:** Effects of the interactions between the incubation duration and temperature and the agar concentration on percentage germination of pollen from two *D. rotundata* genotypes.

Genotype	Incubation Duration	3 h				18 h				Mean
	Agar Concentration (%)	0	0.5	0.75	1	0	0.5	0.75	1	
TDr1621001	15 °C	3.0	18.0	18.0	26.0	4.0	19.0	19.0	25.0	16.5 ^a^
	25 °C	4.0	17.0	24.0	17.0	9.0	17.0	29.0	19.0	17.0 ^a^
	35 °C	6.0	29.0	17.0	21.0	7.0	30.0	18.0	21.0	18.6 ^a^
TDr1655018	15 °C	2.0	10.0	7.0	8.0	6.0	11.0	7.0	8.0	7.4 ^b^
	25 °C	4.0	2.0	2.0	10.0	5.0	6.0	2.0	10.0	5.1 ^bc^
	35 °C	0.0	4.0	3.0	5.0	3.0	6.0	4.0	9.0	4.2 ^c^
	Mean	3.2 ^c^	13.3 ^ab^	11.8 ^b^	14.5 ^ab^	5.7 ^c^	14.8 ^a^	13.2 ^ab^	15.3 ^a^	

*p* < 0.001 for the genotype and the agar concentration; *p* = 0.06 and 0.63 for the incubation duration and temperature, respectively. LSD_0.05_ = 1.5 (genotype); LSD_0.05_ = 2.1 (agar). Means followed by the same letter in the column/row do not differ by the LSD test at 5% *p*-value threshold. Further, 15, 25 and 35 °C represent the three levels of incubation temperature.

**Table 2 plants-10-00795-t002:** Effects of the interactions between the incubation duration and temperature and the agar concentration on percentage germination of pollen from two *D. alata* genotypes.

Genotype	Incubation Duration	3 h				18 h				Mean
	Agar Concentration (%)	0	0.5	0.75	1	0	0.5	0.75	1	
TDa1662006	15 °C	6.0	18.0	5.0	4.0	6.0	20.0	6.0	4.0	8.6 ^ns^
	25 °C	5.0	21.0	26.0	12.0	5.0	22.0	29.0	16.0	17.0 ^ns^
	35 °C	7.0	11.0	12.0	12.0	7.0	19.0	17.0	14.0	12.4 ^ns^
TDa1662010	15 °C	4.0	5.0	4.0	4.0	6.0	10.0	6.0	5.0	5.5 ^ns^
	25 °C	3.0	11.0	6.0	7.0	6.0	12.0	5.0	5.0	6.9 ^ns^
	35 °C	2.0	16.0	10.0	3.0	2.0	14.0	10.0	3.0	7.5 ^ns^
	Mean	4.5 ^ns^	13.7 ^ns^	10.5 ^ns^	7.0 ^ns^	5.3 ^ns^	16.2 ^ns^	12.2 ^ns^	7.8 ^ns^	

Agar (*p* < 0.001); genotype (*p* < 0.001); temperature (*p* = 0.0348). LSD_0.05_ = 4.24 (agar); 2.99 (incubation duration); 3.0 (genotype); 3.67 (temperature). All the interactions’ effects were not significant (ns). Further, 15, 25 and 35 °C represent the three levels of incubation temperature.

**Table 3 plants-10-00795-t003:** Effects of the polyethylene glycol (PEG) on germination percent of yam pollen.

Species	Genotype	Treatment	Pollen Germination (%)
*D. alata*	TDa1662006	Without PEG	45.0 ^a^
		With PEG	15.0 ^b^
	TDa1662010	Without PEG	4.0 ^b^
		With PEG	1.0 ^b^
*D. rotundata*	TDr1621003	Without PEG	45.0 ^ns^
		With PEG	30.0 ^ns^
	TDr1614205	Without PEG	8.0 ^ns^
		With PEG	13.0 ^ns^

ns = no significant differences among the means for *D. rotundata* species. Means followed by the same letter in the column do not differ by the LSD test at a 5% *p*-value threshold.

**Table 4 plants-10-00795-t004:** Hand pollination success vs. in vitro pollen germination.

Species	Pollen Parent	Female Parents	Successful Fruit Set (%)	Pollen Germination (%)	Prediction Accuracy (%) *
*D. rotundata*	TDr1621001	TDr1620009	0.0 (38)	16.8	25.2
		TDr1669010	11.0 (148)		
		TDr1620004	1.7 (57)		
		Mean	4.2 (243)		
	TDr1621012	TDr1601008	31.5 (158)	37.1	64.1
		TDr1620009	18.1 (68)		
		TDr1669010	45.5 (126)		
		TDr1620015	0.0 (17)		
		Mean	23.8 (369)		
	TDr1621003	TDr1620009	16.3 (436)	37.5	32.3
		TDr1621009	9.4 (87)		
		TDr1620029	0.0 (48)		
		TDr1669010	26.9 (26)		
		TDr1620015	20.0 (183)		
		TDr1620004	0.0 (113)		
		Mean	12.1 (893)		
	TDr1614205	TDr1680036	13.3 (117)	10.6	79.7
*D. alata*	TDa1662006	TDa160303	50.7 (311)	27.5	59.7
		TDa1662002	35.1 (233)		
		Mean	46.0 (544)		
	TDa1662010	TDa160303	29.7 (298)	7.7	26.4
		TDa161001	60.6 (19)		
		TDa1662002	32.0 (61)		
		TDa1662003	9.5 (24)		
		TDa1680003	22.6 (416)		
		Mean	29.3 (818)		

* Represents the ratio between the average successful fruit set (%) for a male genotype (observed value) and its percent pollen germination (expected value). The figures in parentheses correspond to the number of flowers pollinated.

**Table 5 plants-10-00795-t005:** Female genotypes used for in vivo fertilization experiment.

Species	Genotypes
*D. alata*	TDa160303
	TDa161001
	TDa1662002
	TDa1662003
	TDa1680003
*D. rotundata*	TDr1601008
	TDr1620004
	TDr1620009
	TDr1620015
	TDr1620029
	TDr1621009
	TDr1669010
	TDr1680036

## Data Availability

All the data is contained within the article.

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
