# Peer review of "Optimized Protocol for In Vitro Pollen Germination in Yam (Dioscorea spp.)"

_plants, 2021, doi:10.3390/plants10040795_

Round 1
Reviewer 1 Report
Dear authors, next to my added word file with comments, please include some of the descriptions from your review in Agriculture 2020. There you show fruits and seeds of yams plus the correct definition of pollen viability. Without knowing that fruits and seeds are (almost) the same in yam, the current data are difficult to understand for someone not working with this species, especially in relation to breeding.

Author Response
Many thanks for your suggestions.
We have intensively made correction and the manuscript have been adjusted.

Reviewer 2 Report
In this study, the authors tested various in vitro pollen germination conditions to optimize conditions for two Dioscorea species. The overall goal is to develop robust in vitro testing conditions for pollen viability to improve breeding in yam. It is well known in the reproduction field that in vitro pollen germination conditions for a species are determined by trial and error. Media that works well in one species might be totally unsuitable for another. The development of a robust and reproducible media and growth conditions would be a benefit to the larger community studying Dioscorea and breeding programs that could use it to accelerate progress in variety improvement.
I do have concerns that need to be addressed.
Why are the values in table 1, 2 and 3 only whole number values (e.g. 3.0%)? References to data in the text include a tens digit (e.g. 31.5%). This leads me into another issue. The description of the statistical analysis is incomplete. There are no sample sizes given (how many pollen were scored for germination and how many replicates were carried out per test)? In figures 2, 3, 4 and 5 some columns are missing error bars. Is this because they are a single replicate, or a typographical error?
I would also prefer a more gradual transition at the start of the results. Given that the methods are at the end of the paper, a brief statement of the overall experimental design at the start of the results would give a better sense of purpose for what follows. I would also appreciate one or more images of germinated pollen.

Author Response
Many thanks for your interest in our research and for the time spent in reviewing it.
We have made correction accordingly.